



# Intraskeletal variability in phosphate oxygen isotope composition reveals regional heterothermies in marine vertebrates.

Nicolas Séon[1], Romain Amiot[2], Guillaume Suan[2], Christophe Lécuyer[2,a], François Fourel[3], Fabien Demaret[4], Arnauld Vinçon-Laugier[2], Sylvain Charbonnier[1] & Peggy Vincent[1]

[1] Centre de Recherche en Paléontologie - Paris (CR2P), CNRS, Muséum national d'Histoire naturelle, Sorbonne Université, 57 rue Cuvier, 75231 Paris Cedex 05

[2] Univ Lyon, UCBL, ENSL, UJM, CNRS, LGL-TPE, F-69622, Villeurbanne, France.

[3] Laboratoire d'Ecologie des Hydrosystèmes Naturels et Anthropisés, CNRS UMR 5023, Université Claude Bernard Lyon 1, Villeurbanne, France

[4] Observatoire PELAGIS, UMS 3462 CNRS/Université de La Rochelle, Pôle Analytique, 5 allée de l'Océan, 17000 La Rochelle, France

[a] Also at Institut Universitaire de France

*Correspondence to*: Nicolas Séon (nicolas.seon@edu.mnhn.fr)

**Abstract.** Strategies used by marine vertebrates to regulate their body temperature can result in local variations, and the knowledge of these regional heterothermies is crucial for better understanding the thermophysiologies of extant and extinct organisms. In order to investigate regional heterothermy in vertebrates, we analysed the oxygen isotope composition of phosphatic skeletal elements ($\delta^{18}O_p$) of two poikilothermic (Atlantic bluefin tuna and swordfish) and three homeothermic endotherms (dolphins). We observed a consistent link between $\delta^{18}O_p$ variations and temperature heterogeneities recorded by classical methods. Our $\delta^{18}O_p$ data indicate that: (i) bone hydroxyapatite of the axial skeleton of dolphins mineralize at a warmer temperature than that of the appendicular one, (ii) the skull is the warmest body region in swordfish, and (iii) Atlantic bluefin tuna possesses high body temperature in the skull and visceral mass region. These results demonstrate the possibility of tracking regional heterothermies in extant marine vertebrates using the $\delta^{18}O_p$, paving the way to direct assessment of thermophysiological specificities of both living and extinct vertebrates. From a paleoenvironmental perspective, the significant observed $\delta^{18}O_p$ variability questions the use of some taxa or random skeletal elements for the reconstruction of paleoceanographic parameters such as seawater temperature and $\delta^{18}O$.

**Keywords:** marine vertebrates, oxygen isotopes, regional heterothermy, thermophysiology, hydroxyapatite.



## 1 Introduction

Within vertebrates, ectotherms (e.g. crocodylomorphs, snakes, lizards, turtles, lissamphibians, chondrichthyans and osteichthyans) rely on environmental heat sources to reach their optimal functional body temperature (Rodbard, 1953) and use behavioural adaptations to maintain it (Crawshaw and Hammel, 1971; Smith, 1979; Hight and Lowe, 2007). Contrarily, endotherms (birds and mammals) produce their body heat physiologically through metabolic processes (e.g. Cannon and Nedergaard, 2004; Legendre and Davesne, 2020). Some of them, the homeotherms, maintain their organs and nervous system

at a nearly constant temperature (within ± 2 °C) by regulating their thermogenesis and thermolysis (Scholander, 1955) while poikilotherms possess deep body temperature which covaries with environmental temperatures (Schmidt-Nielsen et al., 1966; Bennett et al., 1993; Sherwin, 2010; Allali et al., 2013; Nicol, 2017). Maintaining a high and constant temperature throughout the body in non-normothermic conditions is extremely energy-consuming for homeotherms. Consequently, many of them let the temperature of some areas of the body drop to reduce their energy need and limit heat losses (Irving and Hart, 1957;

Rommel et al., 1992; Eichhorn et al., 2011). On the other hand, some poikilotherms are able to produce heat locally (Carey, 1982; Block, 1986; Dickson and Graham, 2004) to improve visual acuity in cold environment (Block, 1987; Fritsches et al., 2005), swim faster or migrate longer distances (Bernal et al., 2001; Blank et al., 2007; Watanabe et al., 2015). These two strategies lead to temperature heterogeneities called regional heterothermies which can be measured on extant organisms thanks to thermometer reading (Ponganis et al., 2003; Morkel et al., 2012) and thermal imagery (Hampton et al., 1971;

Tattersall et al., 2009). However, such methods suffer from several types of limitation. Indeed, *in situ* temperature measurements require the handling of the animal, leading to stress-induced and thus punctual rises in body temperature (Bouwknecht et al., 2007), whereas the infrared thermography is inefficient underwater. It is also difficult to apply them to large and rare living organisms, and in any case impossible to apply on extinct ones.

A possible way to track intra-individual temperature heterogeneities and thus regional heterothermies of both extant and extinct

marine vertebrates could be the use of the oxygen isotope composition of phosphate ($\delta^{18}O_p$) from bioapatite (the mineral forming the bones, teeth and scales of vertebrates). Indeed, vertebrate $\delta^{18}O_p$ values reflect both the oxygen isotope composition of their body water ($\delta^{18}O_{bw}$), stemming from ingested water in osteichthyans or from food for marine mammals (Telfer et al., 1970; Hui, 1981; Ortiz, 2001; Rosen and Worthy, 2018), and their body temperature due to the thermo-dependent oxygen

isotope fractionation between phosphatic tissues and body fluids ($\delta^{18}O_{bw}$) from which they mineralize (Longinelli and Nuti,

1973; Kolodny et al., 1983; Longinelli, 1984; Luz et al., 1984; Lécuyer et al., 2013). Based on these considerations, it is

expected that homeothermic endotherms record homogeneous intra-skeletal $\delta^{18}O_p$ values, whereas in poikilothermic

endotherms, intraskeletal $\delta^{18}O_p$ variability would highlight regional heterothermies. A few studies have investigated the

intraskeletal $\delta^{18}O_p$ variability in some terrestrial and semi-aquatic extant organisms but the relatively reduced number of

samples (n < 10 per individual) of these datasets considerably limits the significance of the $\delta^{18}O_p$ variability (Barrick, 1998;

Stoskopf et al., 2001; Missell, 2004; Coulson et al., 2008; Clauzel et al., 2020). Some palaeontological studies were focused

on the search of regional heterothermies in dinosaurs (Barrick and Showers, 1994, 1995; Barrick et al., 1996, 1998) but the

observed variability in $\delta^{18}O_p$ through the skeleton was difficult to interpret without any present-day isotopic framework and

concrete evidence that the isotopic method works for extant animals possessing regional heterothermies.

In this study, we present new $\delta^{18}O_p$ data obtained from cephalic, axial and appendicular skeletal elements to document the

$\delta^{18}O_p$ variability in selected marine vertebrates with well-documented regional heterothermies and contrasted thermoregulatory

strategies. We compare the obtained $\delta^{18}O_p$ variations with available body temperature measurements obtained from classical

methods and, finally, we discuss the possibility of using this proxy as a tool to identify thermoregulatory strategies and regional

heterothermies of both extant and extinct marine vertebrates.

## 2.       Materials and methods

**2.1.       Sampled specimens**

Five wild specimens belonging to four fully aquatic marine species were studied. They consist of three homeothermic

endotherms (Delphinidae Gray, 1821: two specimens of *Delphinus delphis delphis* Linnaeus, 1758 (M.1162 and MNHN-ZM-

AC-1876-275), one specimen of *Cephalorhynchus commersonii kerguelensis* Robineau, Goodal, Pichler & Baker, 2007

(MNHN-ZM-AC-1983-058)) and two poikilothermic endotherms (Scombridae Rafinesque, 1815: one specimen of *Thunnus*

*thynnus* Linnaeus, 1758; Xiphiidae Swainson, 1839: one specimen of *Xiphias gladius*). All the specimens sampled in our study

are adult. Dolphins specimens were found stranded on the coasts of western France, Kerguelen archipelago and Algeria

(Supplementary material, Table S1), and are curated at the Observatoire des mammifères et oiseaux marins (PELAGIS, France) and at the Museum national d'Histoire naturelle (MNHN, Paris, France), while the swordfish and Atlantic bluefin tuna specimens were obtained from a local fish shop (See supplementary information 1). Between 24 and 44 skeletal elements per

specimen covering all body regions were analysed for their $\delta^{18}O_p$ values. About 50 mg of each skeletal element were ground into a fine powder using either a Dremel$^{TM}$ diamond-head drill or a mortar and pestle. The cortical part of the bone and areas with minimal physical degradation were selected during the sampling process.

## 2.2.      Oxygen isotope analysis of biogenic apatite phosphate

To measure oxygen isotope ratios of biogenic apatite phosphate by gas mass spectrometry techniques, samples were treated according to the wet chemistry protocol described by Crowson et al. (1991) and slightly modified by Lécuyer et al. (2013). The protocol consists of the isolation of phosphate ions ($PO_4^{3-}$) from apatite as silver phosphate crystals ($Ag_3PO_4$). The $Ag_3PO_4$ crystals were filtered, dried and cleaned. For each sample, five aliquots of $300 \pm 20$ µg of $Ag_3PO_4$ were mixed with $400 \pm 50$ µg of graphite in silver foil capsules. Oxygen isotope compositions were measured using a high temperature vario PYRO cube$^{TM}$

elemental analyser (EA) equipped with the "purge and trap" technology (Fourel et al., 2011) and interfaced in continuous flow mode to an IsoPrime$^{TM}$ isotopic ratio mass spectrometer (Elementar UK Ltd Cheadle, UK) at the Plateforme d'Ecologie Isotopique du Laboratoire d'Ecologie des Hydrosystèmes Naturels et Anthropisés (LEHNA, UMR5023, Université Claude Bernard Lyon 1, Lyon, France). Pyrolysis of $Ag_3PO_4$ was performed at 1450 °C. The measurements were calibrated against two standards: a silver phosphate precipitated from the international standards NBS120c (natural Miocene phosphorite from

Florida), and from the NBS127 (barium sulfate precipitated using seawater from Monterey Bay, California, USA). The NBS120c $\delta^{18}O_p$ value was fixed at 21.7 ‰ V-SMOW (Vienna Standard Mean Ocean Water) according to Lécuyer et al.(1993), Chenery et al. (2010) and Halas et al. (2011), and that of NBS127 set at the certified value of 9.3 ‰ V-SMOW (see Hut, 1987; Halas and Szaran, 2001) for correction of instrumental mass fractionation during CO isotopic analysis. Silver phosphate precipitated from standard NBS120c were repeatedly analysed ($\delta^{18}O_p = 21.7 \pm 0.3$ ‰, n = 46) along with the silver phosphate



samples derived from bioapatite to ensure that no isotopic fractionation occurred during the wet chemistry. Data are reported

as $\delta^{18}O$ values normalized to V-SMOW (in ‰ δ units).

### 2.3.    Statistical analyses

To increase sample size and statistical power for testing the intraskeletal variability of $\delta^{18}O_p$ values, skeletal elements were
grouped into several sets corresponding to different parts of the skeleton. The limit between the axial and appendicular skeleton

is set at the articulation between the pectoral girdle and the stylopod for dolphins. For Atlantic bluefin tuna and swordfish, the

fin rays and fin spines belonging to the fins were considered as appendicular skeleton. For the Atlantic bluefin tuna, we have

distinguished anterior and posterior part of the axial skeleton at the limit between precaudal and caudal vertebrae. Since

normality and homoscedasticity of the $\delta^{18}O_p$ values were not validated, we used the non-parametric Mann-Whitney-Wilcoxon
to compare median values between two observational series. Statistical tests were performed using R software (R Core Team,

2017) and the level of significance was set at p-value < 0.05. All the p-values resulting from the statistical tests are reported in

supplementary material, Table S4.

### 3.    Results


The $\delta^{18}O_p$ values of *D. delphis delphis*, *C. commersonii kerguelensis*, *T. thynnus* and *X. gladius* are reported in supplementary

materials, Tables S2 and S3. A synthesis is provided in Table 1. Intraskeletal $\delta^{18}O_p$ variability is represented in Fig.1A for the

north Atlantic *D. delphis delphis* and in Fig.2A and 2B for osteichthyans. The results of the two others Delphinidae studied

are available in supplementary material, Fig. S1. Intra-bone homogeneity was measured by paired samples on vertebrae in
dolphins and osteichthyans and on fin rays in osteichthyans and is systematically lower than inter-bone variability (Table 1).

As expected, the inter-bone $\delta^{18}O_p$ variability is higher in poikilothermic endotherms than homeothermic endotherms (Table

1). For dolphins, $\delta^{18}O_p$ values from the axial skeleton are significantly lower than those of the appendicular ones (p-values <

0.05; Fig. 1B and supplementary material, Fig. S2). Teeth $\delta^{18}O_p$ values of dolphins are higher than those from axial skeletal

(Table 1). Nonetheless, the significance of these differences cannot be tested due to the small number of teeth and skull samples

(n = 1 to 3).  In *T. thynnus*, the highest mean value of 21.6 ± 0.2 ‰ (1SD, n = 6) is recorded in the posterior part of the axial

skeleton, whereas the lowest values (Table 1) are recorded in the skull (20.6 ± 0.5 ‰, 1SD, n = 5) and teeth (20.1 ‰, n = 1).

The skull $\delta^{18}O_p$ values are significantly lower than those of all the other body parts except from those of the anterior part of

the axial skeleton (p-value > 0.05; Fig. 2C). The $\delta^{18}O_p$ values of the skeletal elements belonging to the anterior part of the axial

skeleton are significantly lower than those belonging to the posterior part of the axial skeleton (p-value < 0.05; Fig. 2C). The

mean $\delta^{18}O_p$ value of *X. gladius* whole skeleton is 22.0 ± 0.5 ‰ (1SD, n = 33), with the highest mean $\delta^{18}O_p$ value corresponding

to the rostrum (22.3 ± 0.3 ‰, 1SD, n = 5) and the minimum mean value in the skull (20.7 ± 0.6 ‰, n = 3; Table 1). No

significant differences in $\delta^{18}O_p$ values are observed between either axial skeleton and fins or axial skeleton and rostrum, but

the $\delta^{18}O_p$ values are significantly different between fins and rostrum (p-value < 0.05; Fig. 2C). Despite the small number of

samples from the skull (n = 3), the $\delta^{18}O_p$ values from this body region are lower than all the other ones.

To sum up, phosphate oxygen isotope compositions reveal variations for all studied specimens: the appendicular skeleton in

dolphins is significantly [18]O-enriched compared to the axial skeleton. Swordfish has the lowest $\delta^{18}O_p$ values in the skull and

Atlantic bluefin tuna has the lowest $\delta^{18}O_p$ values in the skull and skeletal elements positioned near the visceral mass.

## 4.    Discussion

### 140  4.1.    Sources of intraskeletal $\delta^{18}O_p$ variability

The measured intraskeletal $\delta^{18}O_p$ variability may results from two main factors identified as the difference in temperature of

bone mineralization across the skeleton as well as changing isotopic compositions of oxygen sources throughout the animal

life. We found significant $\delta^{18}O_p$ differences (~ 0.5‰) between axial and appendicular bones in dolphins that possess the same

mineralization process, strongly suggesting a dominant temperature control (Fig.1B). By contrast, the differences in $\delta^{18}O_p$

recorded between bones and teeth of dolphins (Table 1; Fig.1B and supplementary materials, Fig. S2) cannot be exclusively

attributed to variable body temperature since these elements mineralize at distinct times during ontogeny and possess different

rates of remodelling (Myrick, 1991; Ungar, 2010). For osteichthyans with high metabolic rates such as tunas and billfishes,

mineralization timing should affect $\delta^{18}O_p$ minimally because all skeletal elements are remodelled (Rosenthal, 1963; Atkins et

al., 2014) and teeth are continuously renewed in tunas. The differences in $\delta^{18}O_p$ values between skeletal elements with

comparable timing of mineralization and remodelling rates can therefore be confidently attributed to differences in body

temperature (Fig.2C). Besides, studied organisms are nektonic predators that feed on fish and invertebrates (Young and

Cockcroft, 1994; Kastelein et al., 2000), which in turn possess $\delta^{18}O_{bw}$ values similar to that of their surrounding water (Picard

et al., 1998; Pucéat et al., 2003) but vary depending on the geographical area where they live. The food being the main source

of water in dolphins, the consumption of preys coming from different water masses should cause variations in their $\delta^{18}O_{bw}$.

Nevertheless, the seasonal changes in $\delta^{18}O_{sw}$ of the water masses in which the sampled organisms fed are relatively small

($\pm$ 0.4 ‰; supplementary material, table S5) and cannot fully explain the inter-bone $\delta^{18}O_p$ variability reported herein in

dolphins and osteichthyans.

Therefore, the link between $\delta^{18}O_p$ values and the intra-individual body temperature differences previously documented among

the studied animals strongly suggest that the recorded isotopic variability is mainly due to differences in mineralization

temperature rather than different timing of mineralization.

## 4.2.  $\delta^{18}O_p$ variations linked to regional heterothermies in homeothermic and poikilothermic endotherms

### 4.2.1.  Homeothermic endotherms

Intraskeletal $\delta^{18}O_p$ variability of homeothermic endotherms (mapped in Fig. 1A, and supplementary material, Fig.S1) shows

an isotopic enrichment in the appendicular skeleton relative to the axial one. This indicates a lower mineralization temperature

in these skeletal regions. This observation is consistent with the thermoregulatory strategies used by cetaceans having a trunk

at a nearly constant temperature of 36 $\pm$ 2°C (Morrison, 1962; Hampton et al., 1971), in agreement with their high metabolic

activity (Williams et al., 2001), a thick layer of blubber (Lockyer, 1986; Hashimoto et al., 2015) and counter-current heat

exchangers which limit heat losses at the extremities (Scholander and Schevill, 1955). Counter-current heat exchangers,





defined by a particular spatial arrangement of the cardiovascular system, causes cooling of the blood from the arteries in contact

with the veins and results in body temperature proximodistal gradient (Irving and Hart, 1957). The few information available

for dolphins mentioned body temperature variation of 9°C in the limbs whereas trunk body temperature remains constant

(Tomilin, 1950).

Assuming light seasonal changes in marine mammal $\delta^{18}O_{bw}$ values throughout the seasons and considering the $\delta^{18}O_p$ values

obtained (Table 1), our new isotope data can be used to estimate the temperature differences between limb and trunk in the

sampled dolphins using the phosphate-water temperature scale published by Lécuyer et al. (2013):

T°C = 117.4 – 4.5 ($\delta^{18}O_p$ – $\delta^{18}O_{bw}$)          (Eq.1)

The obtained differences in mineralization temperature are of 2 ± 0.5°C for *D. delphis delphis*, and 1 ± 0.5°C for *C.

commersonii kerguelensis*. In other words, our data show that the mineralization temperature of the bone is about 2°C lower

in the limbs than in the rest of the skeleton in *D. delphis delphis* and 1°C in *C. commersonii kerguelensis*. The estimated

temperature differences are lower than those recorded by classical methods. This difference could be explained by the time

average recorded in the bones. The time record being long, in the order of several years (Rosenthal, 1963; Riccialdelli et al.,

2010; Browning et al., 2014), the estimates inferred from bone $\delta^{18}O_p$ represents long term trend rather than precise temperature

at a specific time and probably mitigate these temperature differences.


### 4.2.2.  Poikilothermic endotherms

Locally high body temperatures have been recorded in several species of tunas (Carey and Lawson, 1973; Graham and

Dickson, 2001) and billfishes (Carey, 1982) using classical methods. Heat in tunas is generated using their red swimming and

extraocular muscles (Guppy et al., 1979) and is retained by counter-current heat exchangers (Block and Finnerty, 1994;

Dickson and Graham, 2004). Unlike most teleosts, tunas have red muscles positioned close to the spine, limiting heat transfer

from the body to the surrounding aquatic medium (Graham and Dickson, 2004). Our $\delta^{18}O_p$ values and their variations across

the body are in agreement with the temperature heterogeneities previously measured by other techniques (e.g. Carey and Teal,

1966; Carey et al., 1971, 1984; Graham and Dickson, 2001), with in particular the lowest $\delta^{18}O_p$ values measured in the skull

and vertebrae near the visceral mass (Table 1 and Fig. 2C). Estimated temperature heterogeneities of tuna assuming slight

seasonal changes in $\delta^{18}O_{bw}$ are of $2 \pm 0.5°C$ between fins and the visceral mass region and $4 \pm 0.5°C$ between fins and teeth

(Fig. 3A). These results are consistent with in situ body temperature measurements which indicate a strong thermal gradient

of 4 to 20°C between core temperature and environmental water (Carey and Teal, 1966; Carey et al., 1971; Carey and Lawson,

1973; Carey et al., 1984). However, the absolute temperature differences inferred from the two methods are difficult to compare

as for dolphins. The high $\delta^{18}O_p$ variability observed in branchial arches can be explained by variable thermal exchanges

between hot blood and cold environmental water.

Swordfish has warm brain and eyes through a unique heater organ associated with the rectus eye muscle (Carey, 1982; Block,

1987) linked to a system of counter-current exchangers and buried in a thick adipose mass that stores the heat produced (Block,

1986, 1991). This mechanism allows the swordfish brain temperature to be 5 °C to 30 °C warmer than the surrounding water

while the rest of its body remains close to water temperature (Carey, 1982, 1990; Schwab, 2002; Stoehr et al., 2018). Our $\delta^{18}O_p$

values and the use of the Eq.(1) indicate that the skull temperature is approximately $7 \pm 0.5°C$ warmer than the rest of the body

which is consistent with the global trend provided by *in situ* temperature measurements (Carey, 1982, 1990; Fritsches et al.,

2005).

### 4.3.        Implications for extant and extinct marine vertebrates

The proposed oxygen isotope thermometry complements conventional approaches and thermal imaging methods. The use of

oxygen isotopes represents a valuable alternative method to access temperature heterogeneities over the body in marine

vertebrates for which logger is difficult to install and operate. Unlike techniques involving surgical implants (Carey and Teal,

1966; Ponganis et al., 2008), isotopic method does not require the handling of living animals, that can punctually increase their

body temperature due to stress (Bouwknecht et al., 2007). Moreover, isotopic thermometry provides internal temperature data

contrary to infrared thermal imagery that limits information to skin temperature (Tattersall et al., 2009; McCafferty et al.,

2015). These results open up new perspectives for thermophysiological studies both on extant organisms that are difficult to

handle (e.g. whales) or which are rare (abyssal organisms), but also on extinct organisms for which only skeleton is available



(e.g. Steller's sea cow, extinct cetaceans, ichthyosaurs, plesiosaurs…). Beyond these (paleo-)biological implications, our results also highlight a major issue concerning the use of random skeletal elements of marine vertebrates (e.g. chondrichthyans

and osteichthyans or cetacean bones and teeth) for the reconstruction of paleoceanographic parameters based on the oxygen isotope composition of bioapatite (e.g. seawater temperatures and $\delta^{18}O_{sw}$ values). Intraskeletal variability resulting from regional heterothermies can lead to overestimate seawater temperature or underestimate $\delta^{18}O_{sw}$ values when applying specific fractionation equations to isolated skeletal elements (Fig. 3A, B). For example, the maximum $\delta^{18}O_p$ difference of 2.8 ‰ measured between two bones of the swordfish can result in an overestimation of 10°C of seawater temperature when applying

the phosphate-water temperature scale of Lécuyer et al. (2013) (Fig. 3A). In the same idea, the maximum $\delta^{18}O_p$ difference of 1.8 ‰ measured between two bones of the North Atlantic short-beaked common dolphin can result in an $\delta^{18}O_{sw}$ underestimation of 1.7 ‰ when applying the fractionation equation published by Ciner et al. (2016; Fig. 3B). It is noteworthy that existing fractionation equations available for chondrichthyans and osteichthyans or cetaceans were established mixing various skeletal elements including axial or appendicular bones and teeth (Longinelli and Nuti, 1973; Kolodny et al., 1983;

Yoshida and Miyazaki, 1991; Lécuyer et al., 2013; Ciner et al., 2016). In order to perform accurate paleoceanographic reconstructions, existing fractionation equations will therefore need to be updated to take into account regional heterothermies.

## 5.     Conclusion

Detailed intraskeletal $\delta^{18}O_p$ mapping allows to document regional heterothermies in marine vertebrates. Calculated $\delta^{18}O_p$-

derived temperatures are consistent with temperature heterogeneities recorded by classical methods. This opens up new perspectives on the determination of the thermoregulatory strategies of present-day marine vertebrates for which conventional methods of body temperature measurements are difficult to apply. This also allows to investigate thermophysiologies of extinct vertebrates since the oxygen isotope composition of hydroxyapatite phosphate can be preserved in the fossil record due to its good resistance to chemical processes that take place during burying and fossilization (e.g. Blake et al., 1997; Lécuyer et al.,

1999; Kral et al., 2021). However, these results highlight the need to update the existing fractionation equations established
for chondrichthyans and osteichthyans or cetaceans as they do not take into account the significant intraskeletal δ18Op variability caused by regional heterothermies.

**Data accessibility.**

Stable oxygen isotope compositions are provided in Excel tables as electronic supplementary materials. Informations about Atlantic bluefin tuna are also mentioned in supplementary materials.

**Funding.**

This work was supported by the ANR-18-CE31-0020 "Oxymore" program (N.S., R.A., G.S., C.L., S.C., P.V.).

**Author contribution.**

P. Vincent, G. Suan, R. Amiot and N. Séon contributed to the study conception and design. Material preparation and data collection were performed by N. Séon, R. Amiot, F. Fourel, F. Demaret, A. Vinçon-Laugier, S. Charbonnier and P. Vincent. Material analysis were performed by N. Séon, R. Amiot and C. Lécuyer,. The first draft of the manuscript was written by N.
Séon, R. Amiot and P. Vincent, and all authors commented on previous versions of the manuscript. All authors read and approved the final manuscript.

**Competing interests.**

The authors declare that they have no conflict of interest.




**Acknowledgments.**

The authors thank C. Lefèvre (MNHN) and W. Dabin (PELAGIS) for providing materials.

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



**Figure caption**

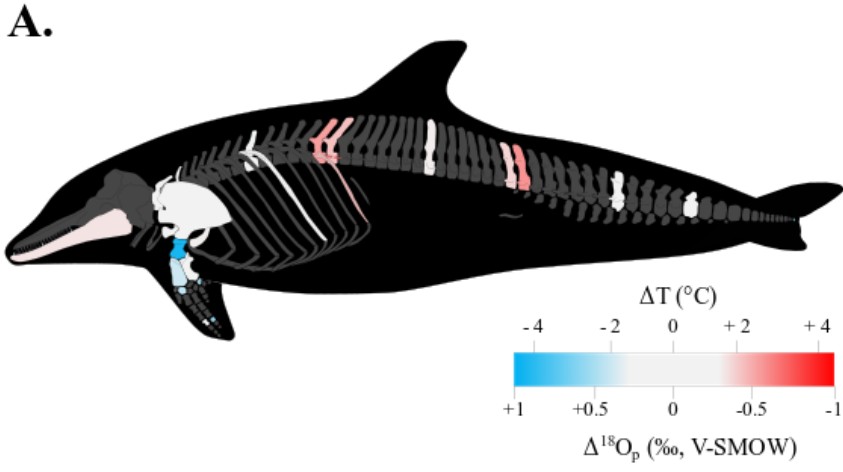

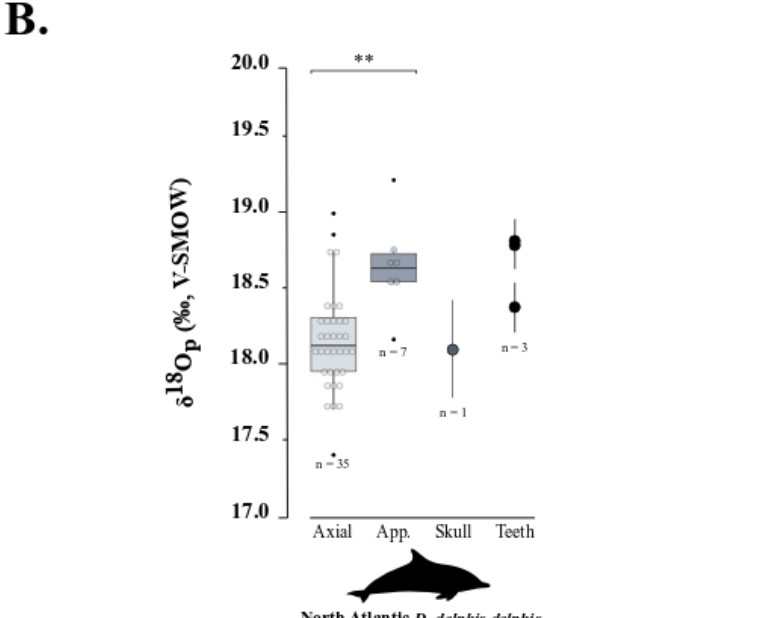


**Figure 1. A.** Oxygen isotope variability within the skeleton of a North Atlantic *D. delphis delphis* (M.1162). Bone Δ18Op correspond to the difference between bone $\delta^{18}O_p$ value and an average value of the skeleton expressed as its mid-range value (($\delta^{18}O_{max}$ - $\delta^{18}O_{min}$)/2). For paired skeletal elements as well as vertebrae centra and neural spines, the mean value is used. **B.** Boxplots showing the $\delta^{18}O_p$ values of skeletal regions for a North Atlantic *D. delphis delphis*. Asterisks indicate the

significance of the observed differences between pairs of groups: ** for $p < 0.01$. Outliers are plotted as small black circles.

Abbreviation = App.: appendicular skeleton.

**Figure 2.** Oxygen isotope variability within the skeleton of *T. thynnus* (**A**) and *X. gladius* (**B**). For each specimen, bone Δ18Op

correspond to the difference between bone $\delta^{18}O_p$ value and an average value of the skeleton expressed as its mid-range value

(($\delta^{18}O_{max}$ - $\delta^{18}O_{min}$)/2). For paired skeletal elements as well as vertebrae centra and neural spines and fin spines and rays, the

mean value is used. Stars represented on the swordfish's skull represents the precise location of the sampling. **C.** Boxplots

showing the $\delta^{18}O_p$ values of skeletal regions for *T. thynnus* and *X. gladius*. Asterisks indicate the significance of the observed

differences between pairs of groups: ns (not significant) for $p > 0.05$, * for $p < 0.05$, ** for $p < 0.01$ (highly significant

difference). Outliers are plotted as small black circles. Abbreviations = Ax.a: axial anterior, Ax.p: axial posterior, Bran.:

branchial arches and Ros.: rostrum.



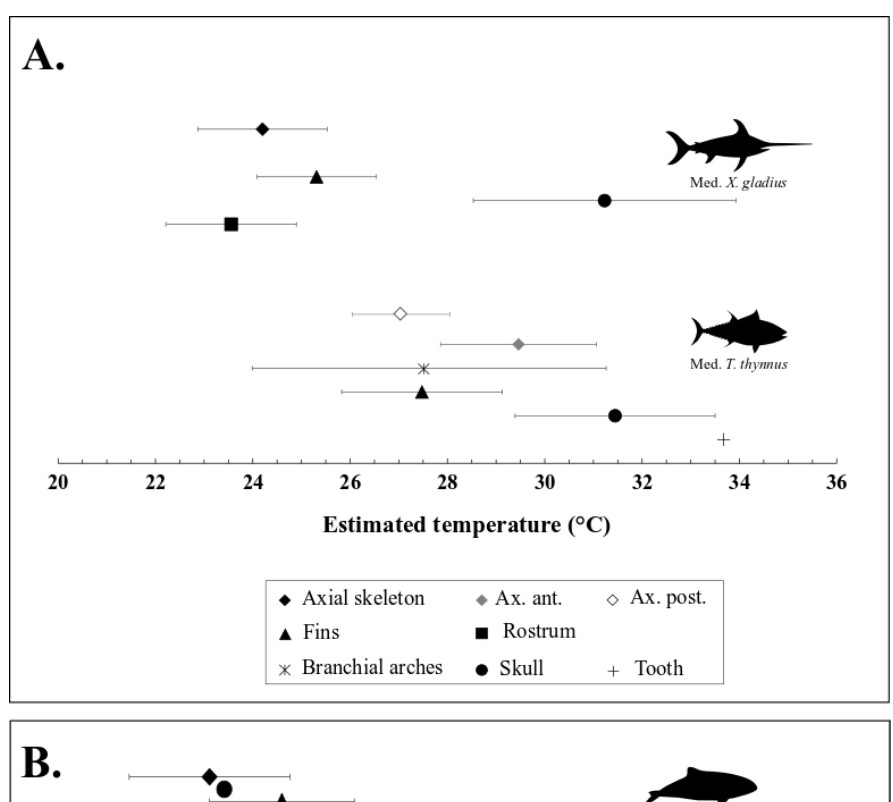

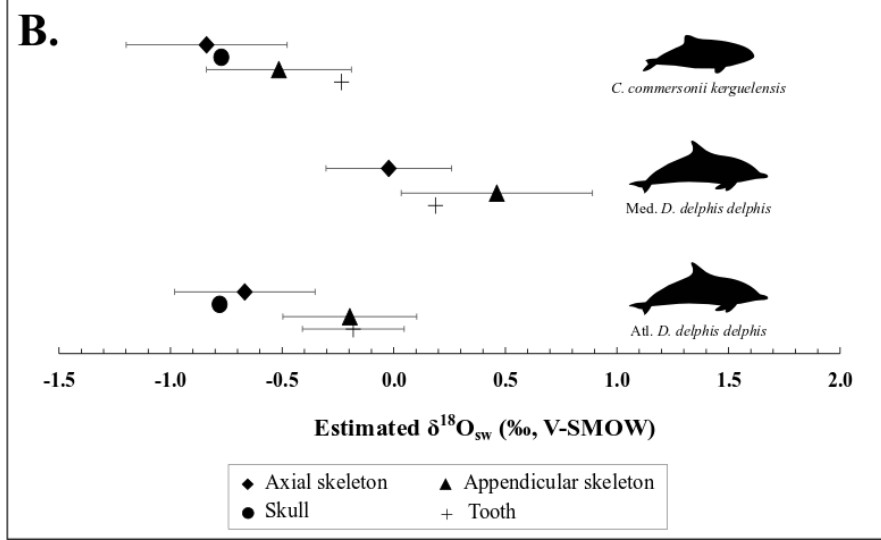

**Figure 3. A.** Estimated hydroxyapatite mineralization temperature from the phosphate-water oxygen fractionation equation

published by Lécuyer et al. (2013), where body water oxygen isotope composition ($\delta^{18}O_{bw}$) for osteichthyans is assumed to be

equals to the $\delta^{18}O_{sw}$ value. Temperature estimates were done with the mean annual oxygen isotope composition of the

Mediterranean Sea ($\delta^{18}O_{sw} = 1.5 \pm 0.4$ ‰; electronic supplementary material, table S5). **B.** Estimated $\delta^{18}O_{sw}$ from the





phosphate-water oxygen fractionation equation published by Ciner et al. (2016). Abbreviations = Med.: Mediterranean sea,

Atl.: Atlantic Ocean.


## Table caption

Table 1. Summary of the mean oxygen isotopic composition (‰, V-SMOW) of dolphins and osteichtyans

| Species | *D. delphis delphis* | | *D. delphis delphis* | | *C. commersonii* | | *T. thynnus* | | *X. gladius* | |
|---|---|---|---|---|---|---|---|---|---|---|
| Inventory number | M.1162 | | MNHN-ZC-AC-1876-275 | | MNHN-ZC-AC-1983-058 | | - | | - | |
| | n | Mean ± SD | n | Mean ± SD | n | Mean ± SD | n | Mean ± SD | n | Mean ± SD |
| Global | 46 | 18.3 ± 0.4 | 33 | 18.9 ± 0.4 | 29 | 18.1 ± 0.4 | 48 | 21.3 ± 0.6 | 33 | 22.0 ± 0.5 |
| Rostrum | | | | | | | | | 5 | 22.3 ± 0.3 |
| Teeth | 3 | 18.7 ± 0.2 | 1 | 19.0 | 1 | 18.6 | 1 | 20.1 | | |
| Skull | 1 | 18.0 | | | 1 | 18.0 | 5 | 20.6 ± 0.5 | 3 | 20.6 ± 0.6 |
| Branchial arches | | | | | | | 6 | 21.4 ± 0.8 | | |
| Axial skeleton | 35 | 18.1 ± 0.3 | 25 | 18.8 ± 0.3 | 19 | 18.0 ± 0.4 | | | 9 | 22.2 ± 0.3 |
|    anterior part | | | | | | | 12 | 21.0 ± 0.5 | | |
|    posterior part | | | | | | | 6 | 21.6 ± 0.2 | | |
| Appendicular skeleton | 7 | 18.7 ± 0.3 | 7 | 19.3 ± 0.4 | 8 | 18.3 ± 0.3 | | | | |
| Fins | | | | | | | 18 | 21.5 ± 0.4 | 16 | 22.0 ± 0.3 |
| $\delta^{18}O_p$ intra-bone variability | 16 | 0.3 ± 0.2 | 7 | 0.5 ± 0.2 | 3 | 0.1 ± 0.1 | 2 | 1.1 ± 0.7 | 4 | 0.4 ± 0.3 |
| Max. $\delta^{18}O_p$* | | 19.2* | | 19.8* | | 19.0* | | 22.5 | | 22.8 |
| Min. $\delta^{18}O_p$* | | 17.4* | | 18.3* | | 17.5* | | 20.0 | | 20.0 |
| Mid-range | | 18.3* | | 19.0* | | 18.2* | | 21.2 | | 21.4 |
| $\Delta\delta^{18}O_p$* | | 1.8* | | 1.5* | | 1.5* | | 2.5 | | 2.8 |

* Teeth are not taking into account in this calculation

**Table 1.** Summary of the mean oxygen isotopic composition (‰, V-SMOW) of dolphins and osteichtyans.