# Peer review of "Intra-skeletal variability in phosphate oxygen isotope composition reveals regional heterothermies in marine vertebrates."

_Biogeosciences, 2022_

## Author Comment (AC1)

Dear Editor,

First of all, we would like to thank the reviewers for the time they spent reviewing our paper and for their constructive comments and suggestions. Comments that will lead to a substantial change in the manuscript are discussed below and all changes to the original text (including minor corrections). Please find below the response to the anonymous referee #1. Most of the corrections requested by the reviewer have taken into consideration and the changes made to the manuscript follow the recommendations of the reviewer. Below you will find the authors' response to reviewer #1 questions and comments.

We hope that these corrections will meet the requirements of the reviewer and editor.

Sincerely yours,

Nicolas Séon for all the authors.

**Response to Anonymous Referee #1**

**General comments**

This paper is welcome because it addresses a technique that has been used to estimate body temperatures of extinct animals from measurements of oxygen isotope in fossil bone and teeth. It recognises that temperature is not uniform throughout the body of aquatic vertebrates (and also terrestrial ones) and shows that isotope analysis of regional bones result in temperatures that are reasonably expected to occur in them. Two groups of extant aquatic species are chosen to represent marine mammals that show regional hypothermy in the limbs and endothermic fish that show regional endothermy by adaptively warming the red muscle, eyes and visceral organs. The results support the use of the method and are strikingly illustrated. The isotope analysis is done according to methods established in the authors' world-class laboratory. The sample sizes are adequate, carefully analysed statistically and interpreted thoughtfully. The writing is generally clear, well organised and extremely well referenced with relevant citations. There are only a few unusual expressions and typographical mistakes that may be rectified by proofreading by a native English writer.

The major problems of the paper involve (1) terminology and (2) use of references. There are suggestions below about terms that may better describe regional patterns of temperature throughout the body. Unfortunately, there appear to be several references that are not used appropriately. Classical references are fine for original ground-breaking research, but more recent papers are best for citing, because presumably they contain the foundational papers as well as recent developments. Some examples are given below.

**Specific comments**

30-37  Some of the citations here and in the rest of the introduction are not very useful or even appropriate. For example, in referencing ectotherms, Rodbard (1953) is a short popular article on 'warm-bloodedness', nearly 70 years old and poorly referenced. The Hight and Lowe (2007) one is more recent, but as it apparently speculates on whether elevated Tb in leopard sharks aggregating in shallow embayments is behavioural thermoregulation. Crawshaw and Hammel (1971) is about Antarctic fish. Norbert Smith's (1979) review of Tb in crocodilians has been long ago superseded (by, for example, G. Grigg and D. Kirshner 2015: Biology and Evolution of Crocodylians). Sherwin (2010) is a manual for animal husbandry and Allali et al. (2013) apparently concerns circadian clocks in camels, both not the

best references for thermal physiology. The introduction would be better if no references were given for generally accepted facts than old specific references that provide little support for the points that the authors are making. Alternatively, more recent reviews on ectothermy and endothermy could be used, for example the Oxford Scholarship 'Ecological and Environmental Physiology Series'. The authors should carefully review the appropriateness and utility of all references in the paper, not only those in the introduction.

As recommended, we have removed inappropriate references or references that are too old and associated with generally accepted facts (Rodbard, 1953; Crawshaw and Hammel, 1971; Smith, 1979; Hight and Lowe, 2007; Schmidt-Nielsen et al., 1966; Bennett et al., 1993; Sherwin, 2010; Allali et al., 2013; Nicol, 2017). We now cite more recent articles in the introduction:

Line 32: Carey et al., 1990

Line 32: McMaster and Downs, 2013

60-63 (also lines 234-235) Here it is essential to provide a citation or two for actual data for regional heterothermy in extant animals.

The following citations were added to line 63: Tomilin, 1950; Carey and Lawson, 1973; Carey, 1982

For lines 234-235,

"Detailed intraskeletal $\delta^{18}O_p$ mapping allows to document regional heterothermies in marine vertebrates. Calculated $\delta^{18}O_p$ derived temperatures are consistent with temperature heterogeneities recorded by classical methods."

Was changed into

"Detailed intraskeletal $\delta^{18}O_p$ mapping allows regional heterothermies in marine vertebrates to be documented. Calculated $\delta^{18}O_p$-derived temperatures are consistent with temperature heterogeneities recorded by classical methods (Tomilin, 1950; Carey, 1982; Graham and Dickson, 2001)" at line 251.

71-74 The groups of animals in the study are classified as either 'homeothermic endotherms' or 'poikilothermic endotherms'. Many would consider the first case misleading, because marine mammals are not wholly homeothermic, but regionally hypothermic, as the authors demonstrate. Also the second case of warm organs in fish seems like a contradiction in terms. 'Poikilotherm' meaning variable temperature is rarely used lately, and for good reason. For example, normal ectothermic fish living in the deep sea cannot be called poikilotherms because their body temperatures are constant. What is important here is that the mammals studied here have cool limbs and the fish have some warm organs. Therefore I would recommend redefining the groups in your study as two types of 'regional heterotherms'. 'Regional' is important to distinguish it from 'temporal heterotherms' which are endotherms that enter hibernation or torpor. In this study, the mammals allow the appendages to cool and the fish warm certain organs. This is simply defined and does not need special titles. You might label sections 4.2.1 'Marine mammals' and 4.2.2 'Endothermic fish'. In any case, avoid the term 'poikilotherm' and its variants.

As requested by the reviewer, the two groups initially named "homeothermic endotherms" and "poikilothermic endotherms" were respectively changed into "Marine mammals" and "Endothermic fish". The term "poikilotherm" is avoided in the rest of the text.

According to the reviewer's comment, the subtitle "4.2. $\delta^{18}O_p$ variations linked to regional heterothermies in homeothermic and poikilothermic endotherms" was replaced by "4.2. $\delta^{18}O_p$ variations linked to regional heterothermies".

144-9   Please explain in greater detail the possible reasons for the differences in predicted temperatures in teeth of fishes and mammals.  One would expect that tuna teeth would form under cold conditions, because tuna respire by ram-ventilation in which the seawater constantly flows in and around their teeth. In contrast, marine mammals have closed mouths. Yet the temperatures derived from isotopes are the opposite to expectations.

The teeth of fish and mammals (especially cetaceans) record different stages of the animal's life. Although tunas breathe by ram-ventilation and therefore in cold conditions, the high efficiency of the *rete mirabile* present near the gills allows a cranial body temperature to be maintained higher than that of the surrounding water (Graham and Dickson, 2001). This explains why we get high mineralization temperature estimates from isotopes. In order to be more precise, we have added the following text to lines 212-215:

"The $\delta^{18}O_p$ values of the teeth indicate that they mineralized at a significantly higher temperature than the fins and the posterior part of the axial skeleton. This is the result of the high efficiency of the *rete mirabile* present near the gills which limits the heat losses associated with ram ventilation (Graham and Dickson, 2001). "

For dolphins, the isotopic composition of the teeth records all the animal life from its development *in utero* to its death. The main part of the tooth is mineralized during the early stages of the animal's development. We have added this following text to lines 153 - 159:

"Indeed, young dolphins breast-feed during the first 12 to 18 months of their life and ingest mother milk that is [18]O-enriched compared to environmental water (Wright and Schwarcz, 1998). Furthermore, odontocetes possess only one generation of teeth that grow a little bit each year until they reach their adult size. It is thus expected that the oxygen isotope composition of teeth is influenced by the [18]O-enriched mother milk unlike bones which are continuously remodelled, thus erasing the isotopic signal of the early animal's development. Due to the small size of the available teeth, we have sampled and analysed the whole teeth; the $\delta^{18}O_p$ values integrate the early stages of the animal's development during which it was breast-feed."

Also, please give a little more detail about remodelling of fish bone. I was not aware that this occurs (=do not know the citations), so please point out how it differs from the amniote paradigm involving secondary osteons.

Bone remodelling was documented in tunas and swordfish (Meunier and Huysseune, 1992; Atkins et al., 2014). What is striking is that bone remodelling occurs in cellular (tunas) and acellular (swordfish rostrum) bones but secondary osteons are present in both. Bone remodelling in fish is still poorly known and needs to be better evaluated and compared to that of mammals (Witten and Huysseune, 2009), which is out of the scope of this paper. From an isotopic point of view, the two processes lead to the incorporation of renewed oxygen. We have therefore added the reference (Meunier and Huysseune, 1992) so the readers can refer to it, but did not add further discussion in the text.

225-231 This final section seems to undermine the whole approach by suggesting the basis for the technique is not up to date.

The first sentence of the final section is indeed misleading and we have changed it to:

"Intra-skeletal variability resulting from regional heterothermies can lead to overestimate seawater temperature or underestimate $\delta^{18}O_{sw}$ values when applying existing fractionation equations that have been established assuming an isotopic homogeneity of the skeleton, to isolated skeletal elements (Fig. 3A, B)."

Please clarify this section by evaluating how much the equations differ and what is the magnitude of the difference.

The sentences of this section (Lines 237 to 244) already document the magnitude of the difference in both marine temperature and $\delta^{18}O_{sw}$ estimates.

Would the study be compromised in its conclusion?

Our study documents and interprets $\delta^{18}O_p$ differences in terms of regional heterothermies, so that our results and their interpretation are not compromised by pre-existing fractionation equations established for cetaceans. Nevertheless, our study draws attention to previous palaeoceanographical studies which did not consider intra-skeletal $\delta^{18}O_p$ variability caused by regional heterothermies. As a result, their palaeoenvironmental reconstructions (seawater temperature and $\delta^{18}O_{sw}$ values) might be inaccurate depending on which skeletal element they analysed and interpreted.

**Technical corrections**

35      Delete 'and thermolysis'. This word means breaking down tissue or cells with heat. It is not appropriate.

"And thermolysis" was removed.

38      Instead of 'non-normothermic conditions is extremely…', I suggest 'at ambient temperatures below the thermal-neutral zone can be…'.

"non-normothermic conditions is extremely…" was changed by "at ambient temperatures below the thermo-neutral zone can be extremely…"

44      Instead of 'thermometer reading' use 'thermometry'.

"Thermometer reading" was changed by "thermometry"

52      I think that both food and water would be taken in by both groups.

Following the reviewer's recommendation, the sentence: "Indeed, vertebrate $\delta^{18}O_p$ values reflect both the oxygen isotope composition of their body water ($\delta^{18}O_{bw}$), stemming from ingested water in osteichthyans or from food for marine mammals (Telfer et al.,1970; Hui, 1981; Ortiz, 2001; Rosen and Worthy, 2018)…"

Was replaced by:

"Indeed, vertebrate $\delta^{18}O_p$ values reflect both the oxygen isotope composition of their body water ($\delta^{18}O_{bw}$), originating from ingested water, food and inhaled dioxygen (Telfer et al., 1970; Hui, 1981; Ortiz, 2001; Rosen and Worthy, 2018)…"

74      See notes above about the use of 'poikilothermic'.

We have replaced the term "poikilothermic" by "endothermic fish"

141      Instead of 'results' use 'result'.

The correction has been made.

154      Use 'prey', which is the pleural.

The correction has been made.

156      Can you provide some numbers for the bone variability to compare with 0.4 for water?

Inter-bone $\delta^{18}O_p$ variability for each group of studied organisms (dolphins and endothermic fish) were added to lines 162-163.

"Nevertheless, the seasonal changes in $\delta^{18}O_{sw}$ of the water masses in which the sampled organisms fed are relatively small (± 0.4 ‰; supplementary material, table S5) and cannot fully explain the inter-bone $\delta^{18}O_p$ variability reported herein in dolphins and osteichthyans."

Was changed into

"Nevertheless, the seasonal changes in $\delta^{18}O_{sw}$ of the water masses in which the sampled marine vertebrates fed are relatively small (± 0.4 ‰; supplementary material, table S5) and cannot fully explain the inter-bone $\delta^{18}O_p$ variability reported herein in dolphins and osteichthyans (respectively 1.5 ‰ and 2.5 ‰)."

155      Use 'appendicular skeleton', not 'these skeletal regions'.

As recommended by the reviewer, we have replaced the expression "these skeletal regions" by "appendicular skeleton".

171      Use 'little', not 'few'.

"Few" was changed by "little".

177     This equation refers to temperature, but the preceding sentence refers to temperature differences. It is confusing.

In order to clarify, we replaced:

"Assuming only slight seasonal changes in marine mammal $\delta^{18}O_{bw}$ values throughout the seasons and considering the $\delta^{18}O_p$ values obtained (Table 1), our new isotope data can be used to estimate the temperature differences between limb and trunk in the sampled dolphins using the phosphate-water temperature scale published by Lécuyer et al. (2013): $T°C = 117.4 – 4.5\ (\delta^{18}O_p – \delta^{18}O_{bw})$          (Eq.1)

The obtained differences in mineralization temperature are of $2 \pm 0.5\ °C$ for *D. delphis delphis*, and $1 \pm 0.5\ °C$ for *C. commersonii kerguelensis*. "

By

"The temperature differences between limb and trunk in the sampled dolphins can be calculated using differences in their $\delta^{18}O_p$ values and the phosphate-water temperature scale published by Lécuyer et al. (2013):

$T°C = 117.4 – 4.5\ (\delta^{18}O_p – \delta^{18}O_{bw})$          (Eq.1)

Assuming only slight seasonal changes in marine mammal $\delta^{18}O_{bw}$ we calculated differences in mineralization temperature between limbs and trunk of $2 \pm 0.5\ °C$ for *D. delphis delphis*, and $1 \pm 0.5\ °C$ for *C. commersonii kerguelensis*."

188     Carey et al. (1984) indicate that heat in tunas is also produced in the viscera.

The reference (Carey et al. 1984) was added to the text.

201     Use 'have' not 'has'.

The correction has been made.

210 and 215    Here and earlier reference is made to thermal imaging.  I would remove this, because it is irrelevant in these cases.

The text part on thermal imagery was removed.

449     Note subscript and superscript errors.

The corrections have been made.

453    It is hard to see the stars.  The red around the eyes is misleading, especially since the eye is warm, but apparently not measured in this study. If the ring could be white, it would not be confused with the red on the temperature scale.

We have replaced the stars by arrows. The rings were measured, that's why it appears in red.

461    Use 'equal', not 'equals'.

The correction has been made.

465    In the footnote to the table, use 'taken' not 'taking'.

The correction has been made.

---

## Author Comment (AC2)

Dear Editor,

First of all, we would like to thank the reviewers for the time they spent reviewing our paper and for their constructive comments and suggestions. Comments that will lead to a substantial change in the manuscript are discussed below and all changes to the original text (including minor corrections). Please find below in blue the response to the anonymous referee #2. Most of the corrections requested by the reviewer have taken into consideration and the changes made to the manuscript follow the recommendations of the reviewer.

We hope that these corrections will meet the requirements of the reviewers and editor.

Sincerely yours,

Nicolas Séon for all the authors.

**Response to Anonymous Referee #2**

**General comments**

The manuscript "Intra-skeletal variability in phosphate oxygen isotope composition reveals regional heterothermies in marine vertebrates" by Séon et al. is an interesting new contribution demonstrating that substantial differences in d18Op values of different skeletal parts exist within ectotherm and endotherm marine vertebrates, which has implications both for temperature and/or salinity reconstructions based on bioapatite phosphate oxygen isotope analysis. The manuscript thus provides notes of caution for such palaeoceanographic seawater temperature and salinity reconstructions which may have a larger error range than previously thought. To support this claim the manuscript presents a convincing and substantial d18Op dataset on modern cetaceans and osteichthyians, it is concise and well written therefore I have only several minor suggestions/corrections to propose.

I miss some information on the salinity and water temperature differences in the method section for the regions from which the marine vertebrates where captured.

To comply with reviewer's request the information concerning salinity and water temperature differences have been added to the supplementary table 5. However, as these informations are not used in the study we did not mention it in the main text.

In the results section you must provide ranges for d18Op values for intra- and inter-bone variability and state that the variability is higher for poikilothermic versus homeothermic endotherms.

In the results section, we have added "In dolphins, the maximum intra-bone $\delta^{18}O_p$ variability (0.5 ‰) is three times smaller than the inter-bone $\delta^{18}O_p$ variability (1.5 ‰; Table 1). In osteichthyans, the intra-bone $\delta^{18}O_p$ variability can reach 1.1 ‰ in *T. thynnus* and 0.4 ‰ in *X. gladius* but still remains lower to the inter-bone variability (2.5 ‰ for *T. thynnus* and 2.8 ‰ for *X. gladius*)."

I think it could be useful to provide a graph and/or text to quantify the influence (error range) of intra-skeletal d18Op variability on water temperature and d18Owater reconstructions.

We have provided (lines 238 to 243) some quantification of the influence of $\delta^{18}O_p$ variability on the temperature and $\delta^{18}O_{sw}$ reconstruction but kept it as a prospect for a future study.

A comparison of measured body temperature differences versus calculated body temperature differences from d18Op values and estimated versus measured d18Osw (when available) might be instructive.

Comparison between measured and calculated body temperatures are provided lines 191, 194, 208-209 and 222, and estimated versus measured $\delta^{18}O_{sw}$ are now illustrated in Fig. 3B and discussed lines 240 to 245.

**Minor comments**

Line 21, 238, 459: hydroxylapatite is the correct terminology according to the IMA (International Mineralogical Association)

The corrections have been made.

Line 36: do you mean core body temperature (instead of deep) here?

We have changed the sentence to "… while poikilotherms possess a core body temperature which covaries with environmental temperatures…".

Line 52: inhaled air oxygen also contributes to the body water pool of lung breathing marine mammals

We thank the reviewer to point this out, it was a mistake and we have changed the sentence as follows "…originating from ingested water, food and inhaled dioxygen…".

Line 56: may be add at the end: in isotope equilibrium

According to the reviewer's comment, we have added "in isotope equilibrium" to the end of the sentence.

Line 58: organisms is to unspecific. Use vertebrates

The correction has been made.

Line 60: paleontological (as you use American English)

The correction has been made.

Line 60: in Vennemann et al. 2001 also intra jaw tooth enameloid d18Op variability of modern sharks is presented

The reference Vennemann et al. (2001) was added to the reference list.

Line 66-68: Would it not be informative to provide a plot at least in the supplements to compare measured and calculated body temperatures (based on d18Op)?

Unfortunately, calculating the absolute measured body temperature with the temperature scale of Lécuyer et al. (2013) is not possible because neither the $\delta^{18}O_{bw}$ or the body water-environmental water $^{18}O$-enrichment is known.

Line 71: four extant fully marine species

The correction has been made.

Line 73: not four authors but Robineau et al. Furthermore, this reference is missing in the reference list

The correction has been made.

Line 76: All three dolphin specimens

The correction has been made.

Line 79: some more provenance information should be provided. Are those specimens form the fish shop from the Mediterranean Sea? Which area?

To be more precise we have changed the sentence in "while the swordfish and Atlantic bluefin tuna specimens were fished in the western Mediterranean Sea".

Line 80: may be you could refer here to one of the figures demonstrating which skeletal parts were sampled

We have added figure references to illustrate which skeletal parts were sampled. "Between 24 and 44 skeletal elements per specimen covering all body regions were analysed for their $\delta^{18}O_p$ values (Fig. 1A, 2A and 2B)."

Line 94, 95, 97: the current terminology for these international standard reference materials is NIST SRM plus the according number

We have replaced NBS 120c by NIST SRM 120c in the corresponding lines.

Line 97: why did you choose a non-matrix matched reference material (BaSO4) and not another isotopically distinct silverphosphate? This is not ideal because of different cumbustion properties of different mineral phases.

As demonstrated in Fourel et al., 2011, the pyrolysis method used in this study (continuous flow technique) generates a single calibration curve regardless of the matrix constituting the reference material. Moreover, we are also using synthetic in-house produced calibrated silver phosphate (Lécuyer

et al., 2019; GGR) for which isotopic compositions are predicted by the thermodynamic properties of the phosphate-water system that were determined by Lécuyer et al. (1999; GCA). Note also that our calibration protocol provides data that are in very good agreement with those obtained during a "ring-test" organised by several worldwide laboratories which results were recently published by Wudarska et al. (2022; GGR).

Line 100: you should state the analytical error of d18Op analysis for samples too or at least mention that it is the same as for NIST SRM 120c.

According to the reviewer's comment, we have added a sentence to state the analytical error of $\delta^{18}O_p$ for samples "Silver phosphate precipitated from standard NIST SRM120c were repeatedly analysed ($\delta^{18}O_p = 21.7 \pm 0.3$ ‰, n = 46) along with the silver phosphate samples derived from bioapatite to ensure that no isotopic fractionation occurred during the wet chemistry. A global analytical error of $\pm 0.3$ ‰ is considered for the whole dataset because the analytical error of the samples $\delta^{18}O_p$ values is smaller or equal to that of NIST SRM120c. Data are reported as $\delta^{18}O_p$ values normalized to V-SMOW (in ‰ $\delta$ units)."

Line 104: intra-skeletal

The correction has been made.

Line 117, 118: Fig. 1A; Fig. 2A (space missing before nr.)

The corrections have been made.

Line 119: you mean variability instead of homogeneity here?

The correction was done with "Intra-bone variability was measured…"

Line 119: why not providing the Fig. S1 in the main text?

We did not provide the Fig. S1 in the main text to avoid the redundancy as the three dolphins display a similar intra-skeletal variability.

Line 120: please provide values for d18Op ranges here

We have added to the main text the values for $\delta^{18}O_p$ ranges: "The $\delta^{18}O_p$ values range from 17.4 ‰ to 19.2 ‰ for the North Atlantic *D. delphis delphis*, from 20.0 ‰ to 22.5 ‰ for *T. thynnus* and from 20.0 ‰ to 22.8 ‰ for *X. gladius*." And the intra-bone variability with ". In dolphins, the maximum intra-bone $\delta^{18}O_p$ variability is three times smaller than the inter-bone $\delta^{18}O_p$ variability (respectively 0.5 ‰ and 1.5 ‰; Table 1). In osteichthyans, the intra-bone $\delta^{18}O_p$ variability can reach 1.1 ‰ but still remains lower to the inter-bone variability (2.5 ‰ for *T. thynnus* and 2.8 ‰ for *X. gladius*; Table 1)."

Line 123: any ideas why the teeth have higher d18Op values? Are the snout regions where they mineralize cooler? The 1.5 permil difference seem to suggest a 6 °C body temperature difference in dolphins. Is this to be expected and in line with instrumental body temperature measurements?

Unfortunately, no snout body temperature data is available for dolphins. The difference between teeth and axial skeleton $\delta^{18}O_p$ values is discussed lines 153 to 159.

Line 141: result (singular not plural)

The correction has been made.

Lines 141-142: what do you mean with oxygen sources of the body: body water, inhaled oxygen?

Can migration to different seawater masses with different d18Osw values play a role here too? What about any mother milk consumption effects? For early ontogenetically forming teeth this could play also a role. Furthermore, could also tissue specific differences in oxygen isotope fractionation (i.e. between dentin and enamel) play any role? Enamel of dolphin teeth is very thin. Thus may be you sampled a mixture between some dentin and enamel.

In order to clarify the concern of the reviewer we inserted the following text in lines 153 to 159: "Indeed, young dolphins breast-feed during the first 12 to 18 months of their life and ingest mother milk that is $^{18}$O-enriched compared to environmental water (Wright and Schwarcz, 1998). Furthermore, odontocetes possess only one generation of teeth that grow at very slow rate each year until they reach their adult size. It is thus expected that the oxygen isotope composition of teeth is influenced by the $^{18}$O-enriched mother milk unlike bones, which are continuously remodelled, thus erasing the isotopic signal of the early animal's development. Due to the small size of the available teeth, we have sampled and analysed the whole teeth; the $\delta^{18}O_p$ values integrate the early stages of the animal's development during which it was breast-feed."

Line 144, 145, 151, 164: space after Fig. missing

The corrections have been made.

Line 149: are not also the teeth of other osteichthyans (not only the tuna) replaced continously? Can you add a reference for this?

We replaced tunas by fish because as mentioned by the reviewer teeth are replaced continuously in osteichthyans. As recommended, we also add two references to justify this point.

"For osteichthyans with high metabolic rates such as tunas and billfishes, mineralization timing should affect $\delta^{18}O_p$ minimally because all skeletal elements are remodelled (Rosenthal, 1963; Atkins et al., 2014) and teeth are continuously renewed in fish (Witten and Huysseune, 2009; Tucker and Fraser, 2014)."

Line 151: Besides, all studied vertebrates…

"Besides, studied organisms are nektonic predators that feed on fishes and invertebrates (Young and Cockcroft, 1994; Kastelein et al., 2000)…"

Was corrected into

"Besides, all studied vertebrates are nektonic predators that feed on fishes and invertebrates (Young and Cockcroft, 1994; Kastelein et al., 2000)…"

Line 151: are different rates of air oxygen inhalation (marine mammals versus fish) not a significant factor for different d18Op values?

Line 151, we did not discuss the $\delta^{18}O_p$ differences between fishes and marine mammals with regard to the different inhaled oxygen levels but it is a very interesting issue. Nevertheless, in this study we did not focus on this factor because it would seem that in marine vertebrates this source of oxygen is not one of the major sources (Hui, 1981; Andersen and Nielsen, 1983; Kohn 1996; Clementz and Koch., 2001).

Line 154: you must quote a reference for the statement that food is the main water source for dolphins.

According to the reviewer's comment, the following sentence "The food being the main source of water in dolphins, the consumption of preys coming from different water masses should cause variations in their $\delta^{18}O_{bw}$."

Was changed into

"The food being the main source of water in dolphins (Telfer et al., 1970; Hui, 1981; Ortiz, 2001; Rosen and Worthy, 2018), the consumption of prey coming from different water masses should cause variations in their $\delta^{18}O_{bw}$."

Line 155: marine vertebrates (instead of organisms)

The correction has been made.

Line 159: is there an estimate possible of how much of the inter-bone variance in d18Op is possible to attribute to temperature differences (based on modelled d18Op from measured temperatures versus measured d18Op)?

These aspects are discussed in the next section 4.2.1 and some temperature differences estimates are provided lines 187 to 194:

"The temperature differences between limb and trunk in the sampled dolphins can be calculated using differences in their $\delta^{18}O_p$ values and the phosphate-water temperature scale published by Lécuyer et al. (2013):

$$T°C = 117.4 - 4.5 (\delta^{18}O_p - \delta^{18}O_{bw}) \qquad (Eq.1)$$

Assuming only slight seasonal changes in marine mammal $\delta^{18}O_{bw}$ we calculated differences in mineralization temperature between limbs and trunk of $2 \pm 0.5$ °C for *D. delphis delphis*, and $1 \pm 0.5$ °C for *C. commersonii kerguelensis*."

Line 164: Intra-skeletal

The correction has been made.

Line 167 and elsewhere in the text: should there not be a space between value and °C ? According to SI unit use guidelines.

This is true! The corrections have been done throughout the text.

Line 167: Is the +/- 2 °C for cetaceans (i.e. dolphins) in line with a +/- 0.5 permil 1 SD variance in measured d18Op? Then worth mentioning this here?

The variability of 2 °C mentioned in the text corresponds to the cetaceans inter-species variability of body temperature not intra-skeletal one.

Line 173-174: no additional, newer references for dolphin body temperature available? What is the constant trunk body temperature, can you provide a value and 1 SD?

We have added a more recent reference: "This observation is consistent with the thermoregulatory strategies used by cetaceans having a trunk at a nearly constant temperature of $36 \pm 2$ °C (Morrison, 1962; Hampton et al., 1971; Yeates et al., 2008),…"

Line 174: Assuming only slight changes…

This sentence was deleted and completely replaced following reviewer 1 recommendation:

"The temperature differences between limb and trunk in the sampled dolphins can be calculated using differences in their $\delta^{18}O_p$ values and the phosphate-water temperature scale published by Lécuyer et al. (2013):

$$T°C = 117.4 - 4.5 \ (\delta^{18}O_p - \delta^{18}O_{bw}) \qquad (Eq.1)$$

Assuming only slight seasonal changes in marine mammal $\delta^{18}O_{bw}$ we calculated differences in mineralization temperature between limbs and trunk of $2 \pm 0.5$ °C for *D. delphis delphis*, and $1 \pm 0.5$ °C for *C. commersonii kerguelensis*."

Line 180-181: could you back this statement up with values how large the differences between reconstructed and measured temperatures are?

According to the reviewer's comment, we have added the differences: "The estimated temperature differences are lower than those recorded by classical methods (respectively 1 °C and up to 9 °C)."

Line 183: … represent a long-term average value…

The correction has been made.

Line 197-200: Is it not possible to compare the temperature variance of measured and calculated temperatures (from d18Op)? Why is the range of core body temperature and ambient water so large (4 to 20 °C)? Because some tuna were caught in cold water settings? I think it would be useful to point this here out as the differences given here based on measured d18Op values are at the lower end of the huge range up to 20 °C quoted.

The core body temperature of tuna, unlike that of cetaceans, is not steady. Indeed, it depends on the activity of their red muscles activity and on the temperature of the surrounding water. Core body temperature increases when the tuna is active, which explain why tunas from the same school can have different core body temperatures. A wide range of temperature from 4 to 20 °C has been measured using classical methods and under various conditions (before, during and after muscle activity).

To clarify, we have added in the lines 209 to 212 the following discussion: "These results are consistent with in situ body temperature measurements which indicate a strong thermal gradient ranging from 4 to 20 °C but most of the time between 5 and 10 °C between core temperature and environmental water depending on both the red muscle activity of the tuna and the temperature of the surrounding water (Carey and Teal, 1966; Carey et al., 1971; Carey and Lawson, 1973; Carey et al., 1984)."

Line 205: Eq. (1), you can refer to Fig. 3A here for the body temperatures.

"Our $\delta^{18}O_p$ values and the use of the Eq.(1) (Fig. 3A) indicate that the skull temperature is approximately $7 \pm 0.5$ °C warmer than the rest of the body which is consistent with the *in situ* temperature measurements (Carey, 1982, 1990; Fritsches et al., 2005)."

Line 206: what do you mean with global trend? Reword for clarification?

Global trend has been removed and rephrased as follow "…warmer than the rest of the body which is consistent with the *in situ* temperature measurements (Carey, 1982, 1990; Fritsches et al., 2005)."

Line 212: …loggers are difficult…

The correction has been made.

Line 217: Well, you need to kill the animal to get bones or teeth for analysis, hence the method is leathal or at least invasive (except for collection of shed teeth or museum specimens). This should be acknowledged. You can add may be… that are difficult to monitor otherwise.  Again, replace the too unspecific organisms by marine vertebrates for which only skeletal remains are available…

"Despite the need of already dead specimens from collections or museums, these results open up new perspectives for thermophysiological studies both on extant organisms that are difficult to monitor (e.g. whales) or which are rare (abyssal organisms), …

Line 218: … and marine reptiles such as... you may additionally mention megatooth sharks.

The correction has been done: "… but also on extinct marine vertebrates for which only the skeleton is available (e.g. Steller's sea cow, extinct cetaceans and marine reptiles such as ichthyosaurs, plesiosaurs…)."

We have chosen not to mention megatooth sharks and sharks more generally given the low potential for preservation of their skeleton.

Line 225: Similarly or Along the same lines, seem more appropriate than the phrasing in the same idea.

"In the same idea, the maximum $\delta^{18}O_p$ difference of 1.8 ‰ measured between two bones of the North Atlantic short-beaked common dolphin can result in an $\delta^{18}O_{sw}$ underestimation of 1.7 ‰ …"

modified in

 "Along the same lines, the maximum $\delta^{18}O_p$ difference of 1.8 ‰ measured between two bones of the North Atlantic short-beaked common dolphin can result in an $\delta^{18}O_{sw}$ underestimation of 1.7 ‰ …"

Line 227: can you please provide the equation you quote here so that the reader is not forced to access the Ciner et al. (2016) reference.

"…when applying the fractionation equation published by Ciner et al. (2016) : $\delta^{18}O_w$ = 0.95317 ($\pm$0.03293) $\delta^{18}O_p$ - 17.971 ($\pm$0.605), r = 0.97253  (Fig. 3B)."

Line 231: would it not be helpful to use the current published equations and illustrate the effects of intra-body d18Op variability on differences in reconstructed d18Osw and body temperature, in supplementary figure(s) for instance?

This will be the topic of another article. We decided not to include it in this manuscript as not to confuse and dilute the main message, which is the possibility of tracking regional heterothermies in marine vertebrates using oxygen isotopes.

Line 239: … chemical alteration processes that take place during postmortem taphonomy and fossilization.

Note that this especially applies for enamel, less so for dentin and bone, which are more liable for alteration (e.g., Ayliffe et al., 1994). Furthermore, it is typically not common practice to quote studies in the conclusion section. I do not know how the Biogeosciences policy is concerning this. If considered ok you can leave as is.

"This also allows to investigate thermophysiologies of extinct vertebrates since the oxygen isotope composition of hydroxyapatite phosphate can be preserved in the fossil record due to its good resistance to chemical processes that take place during burying and fossilization (e.g. Blake et al., 1997; Lécuyer et al., 1999; Kral et al., 2021)."

Changed to

"This also allows to investigate thermophysiologies of extinct vertebrates since the oxygen isotope composition of hydroxylapatite phosphate can be preserved in the fossil record."

Line 241, 441, 449: correct formatting of d18Op (super-, respectively, subscript)

The correction has been made.

**Figures**

Fig. 1B: may be you can provide a typical analytical error bar here?

The analytical error bar was added to Fig. 1B and Fig. 2C.

Fig. 2 may be use same font size as in Fig. 1. May be use same scale for delta d18Op in A and B? The star symbols in B are rather small and difficult to see may be enlarge and fill the stars white to enhance visibility? May be add a note that absolute d18Op differences between the two fish is due to capture in different seawater bodies and mention those.

-The font size in Fig. 2 was modified.

- We prefer to keep the figure this way to highlight intra-skeletal $\delta^{18}O_p$ differences that would disappear in the tuna if we apply the same scale for all.

-We cannot argue that the differences in $\delta^{18}O_p$ are the result of the capture in different seawater masses because we do not have the exact fishing location. We only know that they were caught in the western Mediterranean Sea where the $\delta^{18}O_{sw}$ varies little. We believe that the differences in $\delta^{18}O_p$ values are more likely due to higher body temperature in *T. thynnus* than in *X. gladius*.

Fig. 3: use same symbol size in A and B. Are mean values and 1SD potted in the figures? Please specify.

We have modified the symbols to obtain the same size in A and B. We have also specified that the points represent the means and that the error bars correspond to 1SD.

Line 461: equal (without s)

The correction has been made.

Fig. 3B: may be plot real d18Osw ranges as shaded bars for comparison if there are such values available from the literature or NOA or other seawater d18O database for the regions of vertebrate capture?

The real $\delta^{18}O_{sw}$ ranges were added to the figure as blue shaded pattern.

Is there any reference to support the assumption that osteichthyians have d18Obw = d18Osw that could be cited here?

Lécuyer et al., 2003 ; Pucéat et al., 2003 ; Dera et al., 2009 ; Picard et al., 1998

Are tooth values dentin and enamel mixtures or pur enamel? Not clear. As dolphin enamel is very thing may be the former?

The caption has been modified from "tooth" to "tooth bulk" to clarify the type of the tooth material analysed in this study.

Line 463: fractionation equation for cetaceans by Ciner et al…. Mediterranean Sea

The correction has been made.

**Table 1**

Replace global by all skeletal remains

The correction has been made.

You did not cite the Barrick et al. 1992 whale d18Op paper that also contains cetacean d18Op data of modern whales, why not?

We did not cite the Barrick et al. 1992 paper because we could not use the dataset to test statistically the $\delta^{18}O_p$ differences between teeth and bones. However, we included the reference lines 151-152.

"By contrast, the differences in $\delta^{18}O_p$ recorded between bones and teeth of dolphins (Table 1; Fig. 1B and supplementary materials, Fig. S2) also previously observed by Barrick et al., (1992) and Amiot et al., (2008), cannot be exclusively attributed to variable body temperature because these elements mineralize at distinct times during ontogeny and possess different rates of remodelling (Myrick, 1991; Ungar, 2010)."

---

## Author Response (AR1)

**Author's response**
* * *
Editor's and associate editor's comments in black

Author's responses in blue
* * *
- **18 Jan 2022: Editor's and associate editor's comments**

Comments to the author:

Dear Nicolas and co-authors,

Thank you for your submission to Biogeosciences. Your manuscripts fits well in the scope of BG and I am pleased to have it posted as a pre-print for review. However, I would like to request a few technical changes, listed below - please go through these and upload an updated version, this should not be much work.

Best wishes

Steven Bouillon
* * *
supplementary files:

-I would suggest merging the files into a single pdf

-I'm pleased to see you provide the full data as a supplement. Currently in a pdf file, however, while you mention an Excel format in the manuscript. You should be able to upload an Excel file, or provide the data in a .csv or .txt file if you want to make it more user-friendly for other teams to use your data upon publication (although this can also be done further down the review path if you prefer).

References:

Please check the format - I assume the list was generated with some kind of reference manager software, but it does not include the name of the journals.

https://www.biogeosciences.net/submission.html#references

- **19 Jan 2022: Authors response**

Dear Editor.

Please find attached an uploaded version of the manuscript entitled: "Intraskeletal variability in phosphate oxygen isotope composition reveals regional heterothermies in marine vertebrates". As you will see, most corrections requested by the editor have been applied.

- Supplementary information 1, supplementary figure 1 and 2 were grouped into a single pdf file.

- Supplementary tables 1 to 5 were grouped into a single Excel file. Each supplementary table was reported in an Excel sheet.
- References were corrected. They now include the name of the journals.

Hoping that these corrections will meet with the editor requirements.

Sincerely yours.

Nicolas Séon for all the co-authors
* * *
- **30 Apr 2022: Editor's and associate editor's comments**

Dear Nicolas and co-authors

Thank you for your clear author replies. Based on your initial response to the two review reports - which were both very insightful and constructive, I would welcome a revised version of your manuscript.

Best regards

Steven Bouillon

- **1 May 2022: Authors response**

Dear Editor.

First of all, we would like to thank you for the time you spent reviewing our paper. Please find attached a revised version of the manuscript entitled: "Intra-skeletal variability in phosphate oxygen isotope composition reveals regional heterothermies in marine vertebrates". As you will see, most corrections requested by the editor and the reviewers have been applied.

For more details concerning the corrections, please refer to the authors responses available online.

Hoping that these corrections will meet with the editor and reviewers' requirements.

Sincerely yours.

Nicolas Séon for all the co-authors